# The Influence of Strain Rate Behavior on Laminated Glass Interlayer Types for Cured and Uncured Polymers

**DOI:** 10.3390/polym16060730

**Published:** 2024-03-07

**Authors:** Ahmed Elkilani, Hesham EL-Emam, Alaa Elsisi, Ahmed Elbelbisi, Hani Salim

**Affiliations:** 1Civil and Environmental Engineering, University of Missouri, Columbia, MO 65211, USA; elkilania@missouri.edu (A.E.); elemamh@missouri.edu (H.E.-E.); ahe243@missouri.edu (A.E.); 2Civil Engineering, Southern Illinois University Edwardsville, Edwardsville, IL 62025, USA; aelsisi@siue.edu

**Keywords:** interlayer, laminated glass, high strain rate, cured and uncured polymers

## Abstract

Recent explosions and impact events have highlighted the exposure of civil structures, prompting the need for resilient new constructions and retrofitting of existing ones. Laminated glass panels, particularly in glazed facades, are increasingly used to enhance blast resistance. However, the understanding of glass fragments and their interaction with the interlayer is still incomplete. This paper investigates experimentally the quasi-static and dynamic responses of cured and uncured polymers for seven different materials—two different products of polyvinyl butyral (PVB), two ethylene vinyl acetate products (EVA), one product of thermoplastic polyurethane (TPU), and two SentryGlas products (SG)—that were tested between 21 and 32 °C (69.8 and 89.6 °F), which is the recommended room temperature. In these experiments, the responses of PVB, EVA, TPU, and SG were evaluated under a quasi-static strain rate of 0.033 s^−1^ and compared to the results under a relatively higher strain rate of 2 s^−1^. Moreover, the high strain rate loading of the materials was accomplished using a drop-weight testing appliance to evaluate the engineering stress–strain response under strain rates between 20 and 50 s^−1^. The results demonstrated that with strain rates of 20 s^−1^, PVB behaved like a material with viscoelastic characteristics, but at 45 s^−1^ strain rates, PVB became a non-elastic material. SG, on the other hand, offered both a high stiffness and a high level of transparency, making it a very good alternative to PVB in structural applications. In contrast, after the maximum stress point, the response to the failure of the seven materials differed significantly. The tests provided ample information for evaluating alternative approaches to modeling these different materials in blast events.

## 1. Introduction

During the design of a structure, occupant safety is the most important consideration. A structure’s structural integrity and occupant safety must be considered under external loads, for instance, snow, wind, and extreme events. Storms and explosions can cause extreme damage to structures’ exterior envelopes, making them particularly vulnerable. Several terrorist attacks have taken place in high-density urban areas in recent years, causing building safety to be a concern and a focus on facade design to become more prevalent. In terms of glazing materials for safety, laminated glass (LG) is the most commonly used [1]. Laminated glass is composed of multiple glass layers with a polymer material between them such as polyvinyl butyral (PVB) or ethylene vinyl acetate (EVA). During the failure event, glass fragments adhere to the polymer interlayer and remain stuck [2,3]. It is also important to note that the ductile polymer interlayer deforms extensively after the glass cracks as a continuous membrane. As a result of the interlayer material absorbing energy after breakage, fenestration may remain intact after the blast event, inhibiting air blast waves from entering buildings. Due to its ability to retain most of the sharded glass after failure, laminated glass offers excellent safety benefits when it comes to glass-related injuries. Furthermore, if designed properly, it can prevent air blast waves from entering the building, mitigating hearing-related injuries accompanying blast events [4].

The behavior of LG is primarily dictated by the interlayer type, underscoring its crucial role in the proper design and estimate of structural elements involving LG. Interlayer materials exhibit viscoelastic behavior, implying that their stiffness and mechanical responses vary with the period of the load and the performing temperature [5,6,7,8]. In laminated glass, polymers are usually amorphous and weakly crosslinked which can be used as interlayers. Amorphous polymers, on the other hand, exhibit a certain level of transparency due to the light scattering that occurs at the boundaries between phases, unlike crystalline polymers [9,10,11,12]. There is no latent heat of phase transition in amorphous polymers, which transition from liquid to solid over a long period of time, adopting viscous manners. There is a particular temperature range in which viscoelastic behavior is typical, which is defined by a rubberlike domain, due to its high dependence on temperature. It is common for laminated glass at room temperature for the interlayers to behave like rubber. As a result, when glass breaks, it can form a bridge ligament between the glass shards. There is no way for glass fracture to propagate within a lenient interlayer, rather it departs at the boundary between the glass and the interlayer. This section provides an overview of the materials commonly used in laminated glass interlayers (PVB, EVA, TPU, and SG).

Two pieces of glass are laminated together with a transparent interlayer in PVB laminated glass, by way of windshield in the automotive industry standard. Particularly in passenger cars, PVB laminated windshields are the most widely used windshields [13]. Three different monomers provide specific properties to PVB, which is an amorphous random terpolymer [14]. As PVB elongates several times its initial length over time, it has an extremely time-dependent mechanical response. Several strain magnitudes and strain rates were used to investigate PVB for both cured and uncured properties [15,16]. Previous studies have demonstrated some different parameters of PVB polymer interlayers with 0.03 in (0.76 mm) foil thickness [9]. The PVB water absorption was found to be 3.6% by weight, its coefficient of thermal expansion ranged from 22 to 40 × 10^−5^/°C (71.6 to 104 10^−5^/°F), and its yellowness index was 12.5. Additionally, it was observed that the joining technique with glass can be accomplished in two ways: by lamination and by ultraviolet (UV) curing.

In polyolefins, EVA is a copolymer derived from ethylene and vinyl acetate (VA), with a vinyl acetate weight percentage ranging between 10% and 40%. When VA is polymerized with ethylene, the crystal structure is disrupted, and polyethylene (PE) gets more chemically reactive. Variations in VA concentrations in the composition lead to different properties for EVA. Typically, EVA interlayer materials contain 32–34% VA combined with precisely selected UV absorbers such as additives, curing agents, photo antioxidants, and thermo-antioxidants [17]. Earlier studies have demonstrated some different parameters of PVB polymer interlayers with 0.03 in (0.76 mm) foil thickness [9]. EVA reaches a transmittance of about 90% with a yellowness index of 1.9. As soon as EVA is heated, its coefficient of thermal expansion increases to 160–190 10^−5^/°C (320–374 10^−5^/°F), and its glass transition temperature can be as high as 77 °C (170.6 °F). EVA is a versatile, tough, flexible, elasticity-enhancing, stress-crack-resistance interlayer material. The unique properties of EVA include excellent optical transmission, high electrical conductivity, and low fusion in the autoclave as well. In the photovoltaics (PV) industry, the EVA interlayer is most often used in encapsulating solar energy components [18]. Moreover, EVA interlayer lamination can be performed by vacuum lamination, vacuum bags, or autoclaves.

In the market today, only SentryGlas (SG) based on ionomers, also known as the ionoplast interlayer, is available for laminated glasses [19]. The laminated glass mechanical response has been improved with the use of SG since it was introduced to the market in 1998. Copolymers of ethylene and methacrylic acid are responsible for SentryGlas’ hydrocarbon backbone [18]. A wide temperature range is supported by SentryGlas, which exhibits high stiffness and transparency. Due to its viscoelastic characteristics, it is stiffer than other interlayer materials and has lower sensitivity to load duration and operating temperature than other interlayer materials. For structural laminated glass, SentryGlas provides excellent mechanical properties [9]. Based on the results of this study, all materials were softened by the water immersion environmental effect; SG was especially softened. The coefficient of thermal expansion for SG was found to be 10–15 10^−5^/°C (50–59 10^−5^/°F), and its yellowness index was about 2.5.

All polymers that are based on organic units linked by urethane links are referred to as polyurethane (PU) materials It is important to note that while most polyurethanes are thermosets, interlayer polyurethanes are made of thermoplastic polyurethanes (TPUs), also referred to as polyurethane elastomers. These kinds of polymers are characterized by soft rubbery segments and hard segments that are semicrystalline or glassy. In addition to its excellent tensile strength, TPU is also tough, resistant to UV rays, abrasive, and chemically resistant. TPU material serves as a bridge between rubber and plastic [20]. The strong bonding properties of TPU allow it to be laminated with a variety of materials, including glass, particularly at lower pressures. Therefore, TPU can be applied in hybrid components designed to enhance security and make glass more ballistic resistant.

The most common polymer materials used in safety LG have been studied in several works [21,22,23,24,25,26] for their mechanical properties under static and dynamic strain rates. Consequently, PVB elongates several times its initial length, and its mechanical response is highly time-dependent. According to previous studies [27,28], PVB was tested at a wide range of strain rates. In the field, various window systems were tested with live explosives to examine strain rates at failure. A finite element analysis was conducted [29]. The strain rate of the windows that experienced tearing was 40 s^−1^ at failure; on the other hand, the strain rate of the windows that did not fail was less than 20 s^−1^. 

SentryGlas was tested for tensile performance under different strain rates and displayed elastoplastic behavior [30]. The higher the strain rate, the superior the yield stresses and the greater the amount of elongation. Despite this, the strain rate had no significant impact on failure strength. The final set of tensile tests covered a wide spectrum of strain rates from quasi-static states until 2000 s^−1^, with a rate of 0.0056 s^−1^ for the quasi-static state [31]. Several polymer interlayers were tested at three different strain rates in another study [9], including PVB, SG, EVA, and TPU. Comparisons were made between the mechanical and optical properties of unaged samples and those exposed to thermal cycles, high temperatures, and moisture. As expected, unaged PVB and SG specimens possessed the highest stiffness and tensile strength, while EVA was the most ductile; unaged PVB and SG specimens had the highest tensile strength as well. The aging factors and strain rates of EVA and TPU effects were also less pronounced.

Laminated glass components using EVA exhibit shear transmission better than using PVB. Short-term loadings should be handled with a shear transfer coefficient of 0.7, according to an earlier study [32]. In addition, this study found that laminates with EVA and PVB interlayers had similar ultimate loads. TPU is one of the best interlayer materials on the market. On the other hand, the high cost of TPU prevents it from being widely used nowadays. In addition, TPU interlayers are a very new material, with few certifications, and further research is in progress. An innovative hybrid component was created by combining glass, polycarbonate (PC), and TPU [33]. By combining the properties of laminated glass with the strength of PC, the new hybrid beam has a superior post-breakage behavior compared to laminated glass beams without PC, as well as being lighter and more ductile.

For assessing the long-term stability of modified PVB, thermoplastic polyurethanes (TPUs), ionomers (SGs), and EVA, a previous study used a range of aging scenarios, including temperature storage, climatic stress tests, and tests under high irradiation and aggressive media [34]. The material properties and appearance were both affected by these aging tests on small-scale specimens. After several aging tests, it was found that SG and TPU were the best interlayer materials. Laminated glass interlayers made from these materials can be used both indoors and outdoors for long periods. As a result of the blast load, the LG pane’s glass cracks in the first stage of loading. The laminate interlayer controls the performance of the window after the glass loses its strength. In the process of glass cracking, the interlayer’s performance may be compromised. To that end, this study aims to understand the properties and behavior of PVB, EVA, SG, and TPU interlayers. 

LG resistance is influenced by both environmental and loading conditions. It is important to assess the durability of its mechanical properties against environmental factors to ensure its suitability for safe design applications [35,36,37]. A study investigated the long-term behavior of structural laminated glass plates in a four-point bending test using different interlayers: PVB, EVA, and SG. The experiments were conducted at three temperatures (+20 °C, +30 °C, +40 °C) over 72 h for each interlayer type. Measurements of deflections, longitudinal strains, and displacements were taken. The results indicated that the SG exhibited the lowest deflections, displacements, and strains compared to the PVB and EVA interlayers across different temperature patterns. The difference in endurance performance between the SG and EVA polymeric interlayers was found to be minimal [38]. The fatigue strength of laminated structures has been extensively explored [39,40,41,42]; the cyclic fatigue strength (CFS) of laminated glass with various interlayers and thicknesses was investigated, revealing clear effects of interlayer thickness, type of interlayer, and cyclic bending fatigue load on CFS. Specifically, laminated glass with PVB exhibited lower CFS compared to EVA [43]. 

This paper introduces the experimental data gained from the quasi-static and dynamic loading tests performed for uncured PVB, EVA, SG, and TPU polymeric interlayers and compares it with the cured ones to investigate the curing effect on its mechanical properties, describes the experimental program and proves a strong dependence of their stiffness on the strain rate conditions. Various cured and uncured sheets of PVB and EVA were used to produce tensile test specimens with thicknesses of 0.030 in (0.762 mm), in addition to sheets of 0.035 in (0.889 mm) thicknesses for the SG specimens, while sheets of 0.025 in (0.635 mm) thicknesses were used for the TPU specimens. Testing machines equipped with servo-hydraulics are commonly used for quasi-static testing with strain rates between 10^−5^ and 10^−1^ s^−1^. To test the interlayer materials for high strain, a drop-weight machine was used. Materials with viscoelasticity, such as interlayer materials, have mechanical properties affected by strain rates; therefore, to determine their properties, the strain rates must be determined.

## 2. Methodology and Experimental Program

A description of all experimental designs is provided in this section. A discussion is provided of the preparation of specimens for quasi-static and drop-weight tests. The drop- weight test apparatus, the collection of drop-weight data, and the analysis of drop-weight data are described in detail. An overview of the quasi-static test setup and data acquisition system is provided.

### 2.1. Specimen Preparation

Glass processors commonly use a variety of interlayer materials to produce laminated glass systems. The manufacturers typically receive these materials in sheets with specific thicknesses on rolls. This study used polymer sheets with the following properties as listed in Table 1. Tensile test specimens were created using a range of cured and uncured sheets of PVB and EVA; these sheets had thicknesses of 0.030 in (0.762 mm) for most specimens, except for the SG specimens which used sheets of 0.035 in (0.889 mm) thickness. The TPU specimens, on the other hand, were produced using sheets with a thickness of 0.025 in (0.635 mm).

In this work, the sample geometry for the quasi-static specimens was based on the ASTM D638-10 [44] standard Type IV specimen, as demonstrated in Figure 1a. In addition to the Type I standard specimen, a modified specimen geometry was employed for dynamic testing. In Figure 1b, this modification is shown to have prevented specimens from tearing at the ends of the aluminum tabs, by increasing the bonding area between the aluminum tabs and the interlayer polymer specimens. To ensure precise specimen dimensions, a set of steel cutting dies was carefully manufactured, as shown in Figure 1c,d.

### 2.2. Quasi-Static Testing Setup

Electromechanical static testing frames were used for the quasi-static tensile test. A total of 18 in (457.2 mm) was the distance that this apparatus travels. The device was equipped with load cells with a capacity of 500 pounds (2224 N) and was controlled by a Linear Variable Differential Transformer (LVDT) to measure the sample’s total extension. This apparatus was also equipped with a data acquisition system that transferred the test data to a LabView program which was used to generate the output data. Additionally, the distance between the two grips was also recorded. A high-resolution camera was used to calculate the gauge length deformation. Figure 2a illustrates the quasi-static tension sample that was attached to the grips after putting on the dots to be able to measure the deformation. Additionally, Figure 2c illustrate the parts of the quasi-static servo-hydraulic tensile test machine.

### 2.3. Drop Weight Testing Equipment

The tensile testing at high strain rates was accomplished utilizing a drop-weight testing apparatus. This machine was originally invented for the impact testing of robust materials like metals or composites. For the performance of high-strain rate examinations on polymers, a multitude of novel components were created, produced, and featured. An illustrative diagram of the machine, with critical parts identified, is presented in Figure 3a. The fixed component shown in Figure 3b is a two-segment aluminum plate with dimensions of 14 × 4 × 4 in (355.6 × 101.6 × 101.6 × 101.6 mm). Its chief role is to secure the load cell, which is fastened to the lower part of the fixed component via a threaded connection. This component also serves to halt the weight subsequent to the specimen’s testing. A pair of through holes with a diameter of 1.5 in (38.1 mm) permit the striker’s passage through the fixed component.

Moreover, the movable component, coupled with the forked striker, represents the drop weight as shown in Figure 3c. It is an x-shaped aluminum plate with overall measurements of 14 × 8 × 4 in (355.6 × 203.2 × 101.6 mm), and it weighs a total of 28.7 pounds (127.66 N). This movable component is unrestricted to sliding vertically along the guide rails. Affixed to its lower region is another aluminum plate, to which the forked striker is connected. The movable component is retrieved and elevated to the desired drop height by the lifting mechanism. The striker is composed of two hollow aluminum rods, each 1.5 inches in diameter and 27.5 in (698.5 mm) in length, linked to an aluminum plate measuring 6 × 4 × 1 in (152.4 × 101.6 × 25.4 mm) through threaded connections as seen in Figure 3d. The striker is fastened to the movable component via a bolted connection. The striker’s length allows for an extensive “stroke” length, thus ensuring the rupture of the elastic test materials. Rubber pads, each 1 inch thick, can be placed atop the fixed component to modify the stroke length, acting as a stopper. The striker holds a total weight of 3.4 pounds (15.12 N).

There were two clamps utilized in the drop-weight configuration to secure the test specimen. The upper clamp, illustrated in Figure 3d, was composed of two steel components: one was an L-shaped part directly affixed to the lower region of the load cell through a threaded connection, and the other was a diminutive steel plate measuring 1.25 × 1.25 × 0.5 in (31.75 × 31.75 × 12.7 mm), which formed the clamp’s front face. The lower clamp, or the anvil, was connected to the specimen’s lower end and absorbed the striker’s impact. As this clamp was applied directly to the test specimen, the anvil was constructed with aluminum to reduce the pre-tensioning of the test specimens. The anvil comprised two aluminum plates, each with a thickness of 0.95 in (24.13 mm), linked by a single 5/16 in (7.94 mm) diameter bolt at the center as shown in Figure 3d. The specimens’ elongation was tracked during the testing process using an Edgertronic SC1 model high-speed camera, capturing at a rate of 3000 frames per second. A National Instruments USB-6351 data acquisition system logged the load from the piezoelectric load cell.

### 2.4. Data Processing

The entire event was captured by the high-speed camera. Post-processing methods used camera footage to generate images showing the sample deformation, which could then be used to calculate the sample strain. During loading, gauge length lines were tracked using proprietary software developed in-house. To calculate the engineering strain, the original length was calculated from the frame before loading, as well as the difference in length in all the subsequent frames. As the maximum force was applied, the maximum stress (σ_max_), and strain at failure (ε_f_) were calculated. Moreover, Young’s modulus, E_0_, was evaluated which represents the slope of the first portion of the linear response. Based on the area under the stress–strain curve, Toughness, T, was calculated.

## 3. Experimental Results Interlayer Tests

Specifically, the quasistatic and dynamic stress–strain relations will be discussed in this section, as well as the impact of dynamic strain on the behavior of both the uncured and cured polymer interlayer materials. Sets of stress–strain curves will be parametrized starting with the strain rate effect for each interlayer material, followed by the response of the variable polymers at the same strain rate, i.e., uncured and cured PVB, EVA, SG, and TPU. The last studied parameter will be described in terms of the status of the tested interlayers, i.e., whether they were cured or uncured. All the results from the remaining tests were collected in Table 2.

### 3.1. Interlayer Material’s Strain Rate Effect

Viscoelastic materials, such as polymers, are highly strained and rate-dependent. Therefore, when comparing results, it is significant to ensure the required strain rate is accomplished.

#### 3.1.1. PVB Interlayer

The stress–strain responses of both PVB polymer types (Saflex and Trosifol) at four different strain levels, i.e., quasi-static at 2 in/min (50.8 mm/min) (0.033 s^−1^), high strain at 2 s^−1^, 20 s^−1^, and 45 s^−1^ are shown in Figure 4. Five cured and uncured PVB specimens were cut from a 0.03 in (0.762 mm) thickness rolled PVB sheet by using an aluminum template. The experimental procedure involved conducting tests at varying strain rates until five valid tests were successfully completed. Subsequently, the average values of engineering stress and engineering strain were calculated for the five representative samples. The quasi-static response of both uncured PVB polymer types (Saflex and Trosifol) was hyperbolic in nature and the specimens failed at an average maximum stress of 3250 psi (22.4 MPa) and an average failure strain of 2.6 in/in. The average toughness of the PVB specimens tested was 2600 psi-in/in (17.93 MPa-mm/mm). Moreover, the cured PVB polymer types (Saflex and Trosifol) failed at an average maximum stress of 4355 psi (30.5 MPa) and an average failure strain of 2.0 in/in.

Figure 4 shows the results for both the cured and uncured PVB polymer types (Saflex and Trosifol), which shows that at the dynamic strain rate, PVB started to experience a change in response, especially in the initial region. A higher stiffness was experienced up until a strain of 0.3 in/in. The overall dynamic response was higher than that of the quasi-static response. As the strain rate increased, the dynamic response was approximately bilinear and exhibited a higher stiffness compared to the hyperbolic response under quasi-static loading. Both the cured and uncured specimens failed at maximum stress ranging between 3000 and 4500 psi (20.68 and 31.0 MPa) and a failure strain ranging between 1.6 and 2.5 in/in. The average toughness of the PVB specimens tested was between 2600 and 4500 psi-in/in (18 and 31 MPa-in/in). 

In summary, the differences in the stress–strain behavior between the uncured and cured PVB at varying strain rates were primarily due to the molecular structure of the polymer and the effects of crosslinking on its mechanical properties [3,45]. At low strain rates, such as those typically encountered in standard loading conditions, PVB experienced a viscoelastic characteristic and more elastically, meaning it could deform and return to its original shape when the load was removed [46]. In addition, PVB had sufficient time to move and rearrange in response to the applied stress. This resulted in a more gradual deformation behavior. Unlike the quasi-static response, at very high strain rates or under extreme loading conditions, the material exhibited non-elastic behavior such as plastic deformation or failure. The movement of polymer chains was restricted due to the short duration of the applied stress [47]. This can lead to chain separation, where polymer chains break due to the rapid application of stress, contributing to non-elastic behavior. This transition occurs when the applied stress exceeds the material’s elastic limit; this can lead to a more rapid build-up of stress within the material, causing permanent deformation or fracture.

#### 3.1.2. EVA Interlayer

EVA specimens produced by two different manufacturers were evaluated in this study, namely EVGuard EVA and SE381-TF EVA. Figure 5 illustrates the engineering stress–strain responses of both EVA types at a specific strain rate of 2 in/min (50.8 mm/min) speed for quasi-static as well as 2 s^−1^, 20 s^−1^, and 45 s^−1^ for dynamic responses. Utilizing an aluminum template, five EVA specimens, for each cured and uncured, were cut from 0.03 in (0.762 mm) thick rolled EVA sheets. As part of the experimental procedure, five valid tests were conducted at each strain rate. Afterward, engineering stress and strain values were calculated for these five representative samples. The quasi-static response of both EVA types started with a relatively high initial stiffness followed by a softer response that was hyperbolic in nature. The uncured EVGaurd EVA specimens failed at an average maximum stress of 1250 psi (8.62 MPa) and an average failure strain of 7.3 in/in; the average toughness was 4740 psi-in/in (32.68 MPa-in/in). Furthermore, cured SE381-TF EVA specimens failed at an average maximum stress of 3100 psi (21.37 MPa) and an average failure strain of 5.9 in/in. The average toughness calculated for the uncured EVA specimens was 4660 psi-in/in (32.0 MPa-in/in).

The dynamic stress–strain responses of both EVA types are shown in Figure 5. Initially, both types of EVA responded with relatively high stiffness, followed by a linear softer response. Compared to the quasi-static response, the dynamic response was lower, however, the initial stiffness that all specimens obtained was maintained until a strain of 0.5 in/in. The specimens failed at an average maximum stress of 650–1200 psi (4.48–8.27 MPa) and an average failure strain of 4.5–6.5 in/in. The average toughness was between 2200 and 5500 psi-in/in (15.5 and 37.5 MPa-in/in).

#### 3.1.3. SG Interlayer

Two types of SG polymers were evaluated in this study, namely SG5000 and SG6000. Five SG specimens, from each SG product, were cut using an aluminum template from rolled SG sheets 0.035 in (0.889 mm) thick. In order to test the samples at each strain rate, five valid tests were conducted. The engineering stress and strain values for each of these samples were then determined. Comparisons of the material responses for both SG types at the same previous four strain rates are given in Figure 6, which shows that both SG types exhibited similar behavior. In both SG types, the quasi-static response began with a high initial stiffness, followed by a sudden softening and an almost flat response. Uncured and cured SG5000 specimens failed under the quasi-static strain rate at an average maximum stress of 4400 psi and 6250 psi (30.34 and 43.10 MPa), respectively, as well as the average failure strain of 2.3 in/in and 2.7 in/in, respectively. The average toughness of the uncured SG5000 specimens was 7870 psi-in/in (54.26 MPa-in/in). In addition, both types of SG6000 specimens failed at an average maximum stress of 5930 psi (40.89 MPa) and an average failure strain of 2.6 in/in; the average toughness of the cured SG6000 specimens was 11,200 psi-in/in (77 MPa-in/in).

Based on three different strain rates, Figure 6 illustrates the dynamic stress–strain responses of both SG types. Both types started stiff after being stretched, followed by a softening, and eventually, failure was observed. On average, the SG specimens failed at a peak stress of 6400 psi (44 MPa) and between 1.0 and 2.0 in/in of strain, and their toughness ranged from 5000 to 13,000 psi-in/in (34.47 to 90 MPa-in/in). Compared to the static response, the high strain rate response of SG polymers exhibited a reduction in strain to failure, however, the dynamic maximum stress increased to more than 60% compared to the static response.

#### 3.1.4. TPU Interlayer

The average engineering stress–strain response of the TPU specimens is shown in Figure 7. The engineering stress and strain values for each sample were determined after five valid tests were conducted using rolled TPU sheets of 0.025 in (0.635 mm) thick. In the event of the quasi-static strain rate of 2 in/min (50.8 mm/min) (0.033 s^−1^), the response was hyperbolic in nature with an initial linear segment. A corresponding average failure strain of 4.7 in/in was observed for all the TPU specimens at an average peak stress of 8500 psi (58.60 MPa). The average toughness of the TPU specimens tested was 11,360 psi-in/in (78.32 MPa-in/in). In Figure 7, the dynamic stress–strain response of the TPU specimens is depicted. In contrast to the quasi-static hyperbolic responses, the response to the TPU specimens was approximately bilinear. The specimens failed at 4000 psi (27.58 MPa) and about 3.5 in/in strain. The average toughness of the TPU specimens tested was about 8000 psi-in/in (55.16 MPa-in/in). Under dynamic testing, TPU had significantly lower stress at failure and lower ductility. For both types, the failure strains were significantly lower than those exhibited under a static load.

### 3.2. Interlayer Material’s Types Effect

This study focused on the quasi-static and dynamic response of interlayer polymer materials at room temperature. Preliminary results are also presented in this work for the material response under moderate, low, and high dynamic strain rates.

#### 3.2.1. Quasi-Static Strain Rate

Engineering stress–strain comparisons under a quasi-static loading rate of 2 in/min (50.8 mm/min) (0.033 s^−1^) across different polymer materials for both uncured and cured polymers are illustrated in Figure 8a,b, respectively. Except for SG, the polymers exhibited a complete or partial hyperbolic response to failure. SG had a relatively very high initial stiffness followed by softening and a plastic, relatively flat plateau until strain hardening occurred leading to failure. PVB exhibited a higher strength but a lower failure strain, resulting in toughness for PVB being about half that of the EVA polymers. TPU outperformed both the PVB and EVA polymers. Although TPU had a lower failure strain than EVA, TPU had a much higher strength resulting in a toughness that was about five times that of PVB and two and a half times that of EVA. The static toughness of TPU, which is a good measure of strength and ductility, was comparable to that of the SG polymers. TPU and SG generally experienced the highest stress at failure, whereas EVA experienced the largest strain at failure. Furthermore, at a quasi-static strain rate, TPU and SG exhibited the largest toughness of all the polymers tested. For comparative purposes, if one assumed for design a maximum strain in the polymer interlayer not to exceed 2 in/in, then SG outperformed all the polymers by providing the highest strength and toughness at that specific strain limit of 2 in/in, see Figure 9. 

The variations in the quasi-static stress–strain responses among the tested polymeric interlayer materials could be attributed to their physical and chemical compositions [9]. For instance, EVA offers stress-crack endurance, great flexibility, toughness, and elasticity compared to PVB. These fundamental behaviors can undergo significant enhancement through crosslinking in its chemical structure, resulting in improved resistance to creep rupture, tearing, and chemicals. Whereas, PVB chains with higher molecular weight provide higher mechanical strength compared to EVA, achieved by incorporating plasticizers that enhance the material’s mechanical properties. Through the use of metal ions such as physical crosslinking points, SG achieves the highest levels of stiffness, which categorizes it as a thermoplastic material. TPU’s physical and mechanical behavior, such as high tensile strength and toughness, is determined by its structure, which is influenced by the ratio of diisocyanate, polyol, and chain extender, as well as the chemical reaction conditions.

#### 3.2.2. Dynamic Strain Rate

The dynamic responses of the polymers evaluated in this research were compared to each other at a high strain rate of 2 s^−1^, see Figure 10. SG, followed by TPU, generally experienced the highest stress levels, whereas EVA experienced the largest strains at failure. SG and TPU exhibited the largest toughness of all the polymers tested dynamically. EVA had the least strength of all the polymers evaluated under dynamic loading but had the largest ductility. The results demonstrated that SG had a similar failure strain response to that of PVB, which was much less than other polymers. Overall, the SG polymers experienced the most improvement in terms of maximum stress and toughness (area under the stress–strain curve), however, it provided the least strain to failure. A similar conclusion was obtained from the quasi-static testing that was performed earlier.

As a dynamic strain rate of 20 s^−1^ was applied, Figure 11 illustrates the engineering stress–strain for all the interlayers. In comparison to the other materials, EVA had the lowest strength, while both types of SG had the highest. A maximum yield stress of approximately 6500 psi (44.82 MPa) was observed in the SG interlayers based on their dynamic response. After the yield stress for both types of SG, there was a sudden softening behavior. Due to the SG material’s high brittleness, the softening could be caused by surface cracking. Immediately after the initial yield strength, the stress began to decrease significantly as a result of this softening behavior. The other polymers showed a plastic and/or hyper-elastic response. Among the tested types of dynamic responses, EVA experienced the largest strain at failure, which resulted in greater ductility. It was observed that TPU experienced higher elongation and strength than PVB.

Figure 12 shows the dynamic response of the polymers at a high strain rate of 45 s^−1^. These results established that all the PVB, SG, and TPU types exhibited a superior material response and energy absorption capability compared to the two types of EVA evaluated. The increase in strain rate had minimal impact on the responses of EVA compared to the other interlayer materials. Moreover, the SG interlayer exhibited higher strength, and consequently absorbed higher energy than the other interlayers over the whole dynamic response to the failure of the polymers. Conversely, SG was the hardest interlayer as it had the lowest ductility and elongation. PVB’s strength under dynamic loads was lower than SG, but it was higher than the other polymers evaluated.

The unique properties of each polymeric interlayer played a crucial role in verifying their performance under the tested conditions [9]. TPU polymers consist of block copolymers with rubbery soft segments and semi-crystalline which are characterized by their high elasticity and viscoelasticity. For SG, in the dynamic resistance, it showed the highest stiffness over the tested polymers. The role of metal ions as physical crosslinking points in the polymer chain could potentially explain this phenomenon. On the other hand, PVB’s high elasticity and toughness make it suitable for absorbing energy during dynamic events, such as blasts or impacts. Its ability to maintain adhesion to glass after deformation helps prevent catastrophic failure due to it consisting of three different monomers, contributing specific properties to form an amorphous random terpolymer [48]. Moreover, EVA exhibited the highest dynamic elongation among the tested polymers due to its copolymer composition of ethylene and vinyl acetate. In this case, the quasi-static test results cannot be interpreted as a representative of the material response when subjected to high strain rate loading scenarios, such as those resulting from blast loading from an explosion. Furthermore, the strain rate significantly impacted the initial stiffness and dynamic energy of the PVB and EVA interlayers.

### 3.3. Interlayers Curing Effect

As mentioned previously, the earlier sections compared the stress–strain relationship overall. As part of this section, bar charts are also used to summarize the quasi-static and dynamic testing results of the polymers, comparing the failure stress and strain, elastic moduli, and toughness of the polymers under four different strain rates: 0.033 s^−1^, 2 s^−1^, 20 s^−1^, and 45 s^−1^.

#### 3.3.1. PVB Interlayer

Figure 13 and Figure 14 show a comparison between the maximum failure stress and strain, elastic moduli, and toughness for both PVB types under different strain rates. It follows that as the strain increases, the stress will increase again until the failure occurs. Consequently, all the cured and uncured materials used were sensitive to high strain rates. In both the static and dynamic cases, the curing process resulted in the hardening of the PVB interlayers. The curing process caused an increase of about 36% in maximum failure stress for both types of PVB. Despite this, the failure strain was reduced in both PVB types. In this study, toughness was quantified by the area under the stress–strain curve up to failure, so the curing process increased the toughness by two, especially for PVB-Trosifol. The dynamic response demonstrated a higher stiffness in comparison to the hyperbolic response when the strain rate increased. Curing the PVB-Trosifol polymers significantly increased their initial stiffness, as illustrated in Figure 14c. Additionally, curing greatly increased the elastic moduli of both PVB types, resulting in a 150% stiffer material than the uncured material. Static behavior was more affected by the curing process than dynamic behavior.

#### 3.3.2. EVA Interlayer

Both EVA types were compared under different strain rates as shown in Figure 15 and Figure 16. According to the results, the curing process increased failure stress significantly, particularly for static responses. Based on the comparison between the cured and uncured charts, it was clear that the curing process reduced both failure strains for the EVA interlayers. In consequence, the material’s toughness up to failure decreased, leading to a reduction in energy absorption, while the failure stress increased. Conversely, under a dynamic strain rate of 45 s^−1^, the cured EVGuard material exhibited about double the toughness. For both the static and dynamic cases of EVA materials, the curing process caused a hardening of the materials, except for the dynamic curves of EVA and EVGuard at strain rates of 45 s^−1^ and 20 s^−1^, respectively.

#### 3.3.3. SG Interlayer

In this part, the uncured and cured SG polymers were compared based on their failure stress and strain, elastic moduli, and toughness (see Figure 17 and Figure 18). Unlike the PVB and EVA materials, the curing process had no effect on the maximum failure stress except for the static behavior for the SG5000 and the dynamic behavior for the SG6000 at a strain rate of 2 s^−1^. Meanwhile, the curing process increased the material’s failure strain, which, in turn, raised the material toughness and its ability to absorb more energy. The SG5000 and SG6000, for instance, increased their toughness by 200% and 85%, respectively, after curing. For the SG5000, the initial modulus increased considerably with the curing process; in particular, it reached about 62360 psi (430 MPa) at a dynamic strain rate of 45 s^−1^. In contrast, the curing process did not alter the SG6000’s initial moduli, for instance, it lost 25% of its stiffness at a strain rate of 2 s^−1^. Finally, the results showed both types of SG materials to have maximum stiffness and toughness, as well as the SG material exhibiting a higher brittleness than the other materials. Laminated glass’ ability to absorb energy is one of its most important characteristics, so an increase in toughness at high strain rates is desirable, particularly for blast scenarios. Additionally, there is a very distinct pseudo-yield point between the SG6000 and SG5000 that was characterized by high strength, high strain energy, and a high initial modulus.

#### 3.3.4. TPU Interlayer

The maximum failure stress and strain, toughness, and elastic moduli for both TPU types were compared under different strain rates as shown in Figure 19. Considering the bar charts for the TPU polymers, the curing process increased failure stress for static responses as well as at a strain rate of 2 s^−1^. TPU failure strain was not significantly affected by curing, though a strain rate of 45 s^−1^ showed a 14% difference. These results showed that cured TPU exhibited a superior material response and energy absorption capability compared to the uncured one, except at a strain rate of 20 s^−1^ where toughness decreased. At a specific strain value, the toughness decreased under strain rates higher than 0.033 s^−1^. With regard to the elastic modulus, the curing process increased the stiffness and hardness of the material, especially at a strain rate of 2 s^−1^, where it increased by 33%. In general, TPU and SG exhibited the largest toughness of all polymers tested at a quasi-static and dynamic strain rate.

In summary, the various polymeric interlayers discussed in this work find application across different industries [9]. PVB, for example, is commonly used in laminated glass for applications such as automotive windshields and architectural glazing, providing safety by holding glass fragments together when broken. EVA, on the other hand, is utilized in solar panels for encapsulation, offering adhesion and UV resistance. SG is preferred for high-end architectural glazing due to its strength and clarity, often used in facades and hurricane-resistant windows. TPU finds applications in flexible films for inflatable structures and medical devices, known for its elasticity and abrasion resistance.

### 3.4. Modes of Failure

Based on the test results, it is evident that there are different types of failure modes among the four materials, as shown in Figure 20. PVB and TPU were uniformly deformed over the gauge length of the sample until it was cut in the middle. Conversely, TPU exhibited much greater elongation than PVB. Due to the excessive strain, the straight area of the specimen looked white for EVA, which stretched much more than PVB, TPU, or SG. In the case of SG, it was observed that the stress instantly decreased after the yielding occurred, resulting in a strain localization in the middle of the specimen. Throughout the test, the strain remained localized in the same area. Since the surface of the SG appeared brittle, unlike EVA, TPU, and PVB, the reduction might have been caused by surface cracking after the yield stress.

The experimental results obtained from investigating the quasi-static and dynamic responses of seven different materials, including two variants of PVB, two products EVA, one type of TPU, and two products (SG), presented a significant opportunity to gain insightful understanding into the behaviors of these polymers. By thoroughly examining the cured and uncured states, the experiments provided a comprehensive insight into how these polymers respond under varying conditions. These experimental findings are crucial for correlation with analytical models, contributing to the calibration of material models essential for numerical simulations. This integration of experimental and analytical approaches is pivotal for enhancing the accuracy and reliability of predictions in structural behavior. Moreover, the extensive dataset generated from these experiments serves as a foundation for the development of an artificial intelligence (AI) program. This program aims to predict the behavior of polymer materials across different strain rates and temperatures, paving the way for advanced simulations and predictive modeling in the realm of materials science and engineering.

## 4. Conclusions

To ensure the safety of building occupants, finding ways to improve the integrity of the building façade, especially those that may be subjected to blasts, has become an important area of study for engineers. Understanding the material response of the polymer interlayer used in laminated glass under quasi-static and dynamic loading is critical for developing blast analysis and design methodology. This study focused on the static and dynamic responses of four different interlayer polymer materials, PVB, EVA, TPU, and SG, from two different suppliers tested at room temperature. Based on the results of this work, the following observations are made:Emerging polymers such as TPU and SG showed the largest strength under static and dynamic loads, but that strength was drastically affected under moderate dynamic strain rates.The dynamic toughness of SG, which is a good measure of strength and ductility, was the highest of the polymers evaluated. TPU’s toughness under dynamic loads was lower than SG, but it was higher than the other polymers evaluated.PVB experienced the most change in response from quasi-static to dynamic rates of loading compared to the other polymers. Also, PVB seemed to improve in all areas under dynamic loads compared to the other polymers.The SG stress–strain response was different from the other polymers. It mimicked a ductile material behavior with high linear initial stiffness, yield point, plastic necking, and softening, followed by strain hardening. Other polymers showed a plastic and/or hyper-elastic response.The SG polymers experienced noticeable improvements in their response when loaded under high strain loading compared to quasi-static loading. EVA and TPU experienced the least change to their quasi-static responses compared to high strain rate responses.With increasing strain rate, the stiffening performance of polymer interlayers was clearly noticeable in the SG tests. One remarkable visual examination from these tests was that SG experienced necking throughout part of the gauge length.EVA had the least strength of all the polymers evaluated under static and dynamic loading but had the largest ductility. This combination resulted in EVA having a higher toughness than PVB.SG had a similar failure strain response to that of PVB, which was much less than the other polymers, but had higher strength at lower strain values compared to all the other polymers.At a set limit on strain of 2 in/in, SG outperformed the other polymers. This indicated that for design, SG could provide the best option provided strain levels are kept at a low range.The response of most interlayer polymer materials was significantly altered under dynamic strain rates.

## Figures and Tables

**Figure 1 polymers-16-00730-f001:**
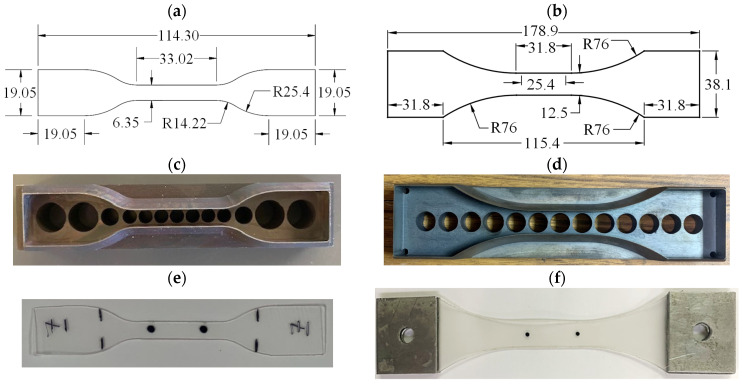
Specimen preparation: (**a**) quasi-static specimen geometry, (**b**) dynamic specimen geometry, (**c**) quasi-static cutting die, (**d**) dynamic cutting die, (**e**) quasi-static sample marking, and (**f**) dynamic sample marking. (Dimensions mm).

**Figure 2 polymers-16-00730-f002:**
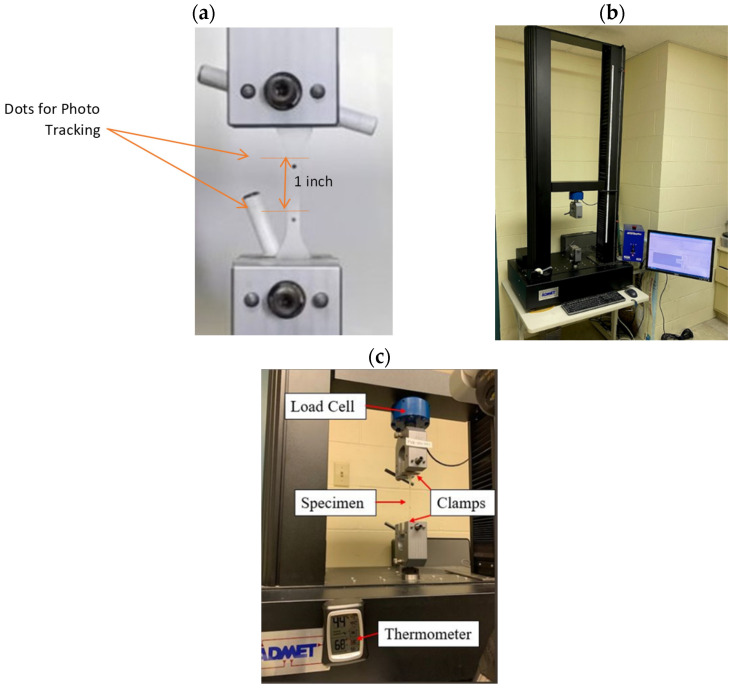
Quasi-static tensile test machine: (**a**) quasi-static tension specimens set for testing, (**b**) servo-hydraulic quasi-static tensile test machine, and (**c**) annotation of quasi-static tensile test machine parts.

**Figure 3 polymers-16-00730-f003:**
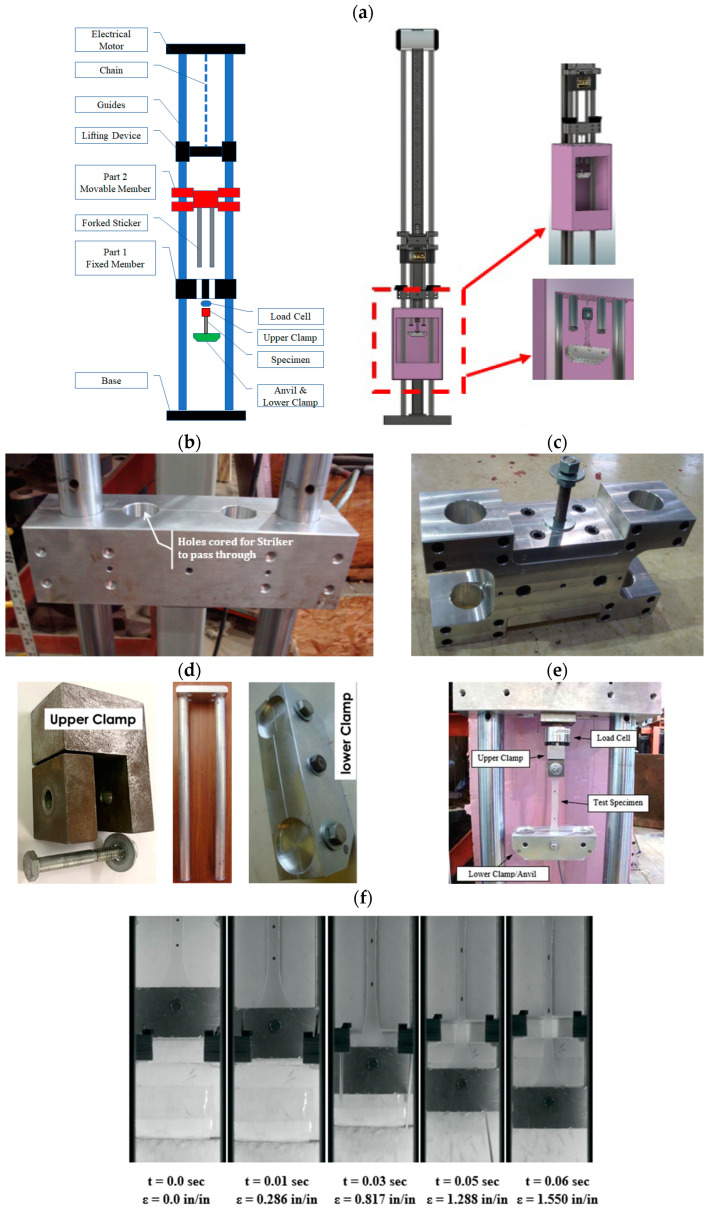
Drop weight test set-up: (**a**) drop weight frame schematic, (**b**) part 1: fixed member, (**c**) part 2: movable member, (**d**) anvil and striker, (**e**) specimen set-up, and (**f**) selected frames from a high-speed video of a PVB test.

**Figure 4 polymers-16-00730-f004:**
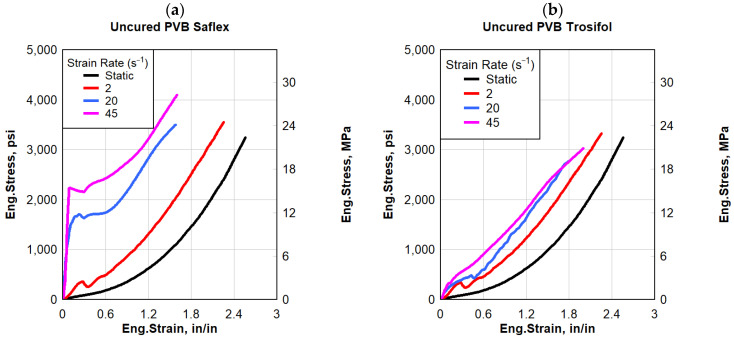
Eng. stress vs. eng. strain: (**a**) uncured PVB Saflex, (**b**) uncured PVB Trosifol, (**c**) cured PVB Saflex, and (**d**) cured PVB Trosifol.

**Figure 5 polymers-16-00730-f005:**
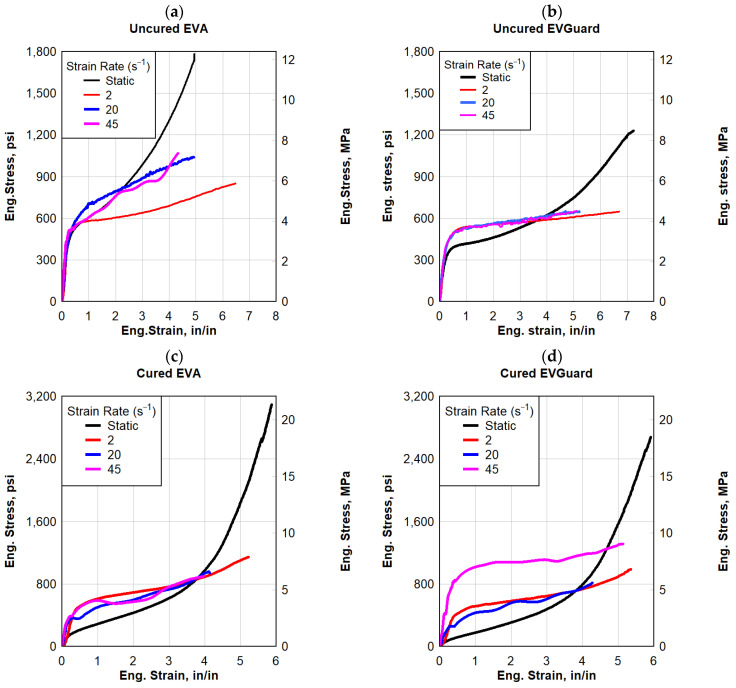
Eng. stress vs. eng. strain: (**a**) uncured EVA, (**b**) uncured EVGurad, (**c**) cured EVA, and (**d**) cured EVGuard.

**Figure 6 polymers-16-00730-f006:**
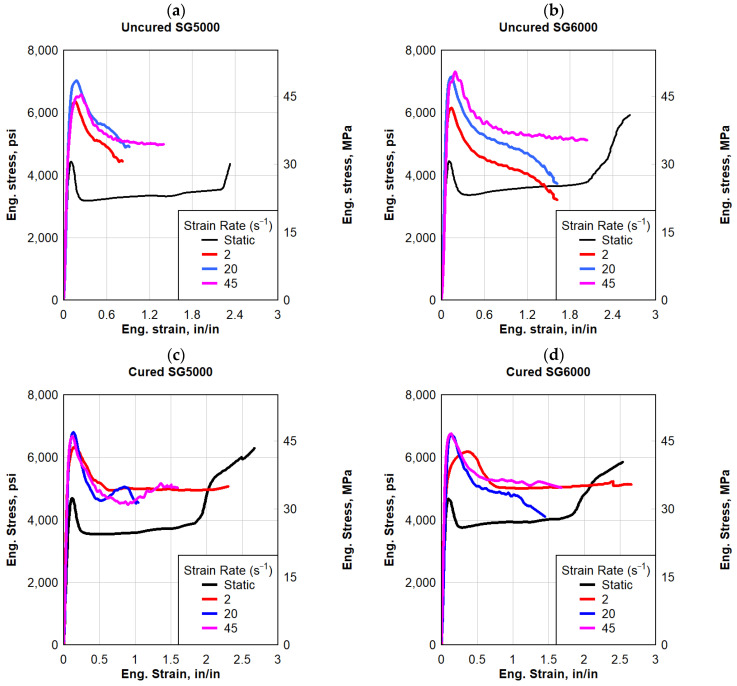
Eng. stress vs. eng. strain: (**a**) uncured SG5000, (**b**) uncured SG6000, (**c**) cured SG5000, and (**d**) cured SG6000.

**Figure 7 polymers-16-00730-f007:**
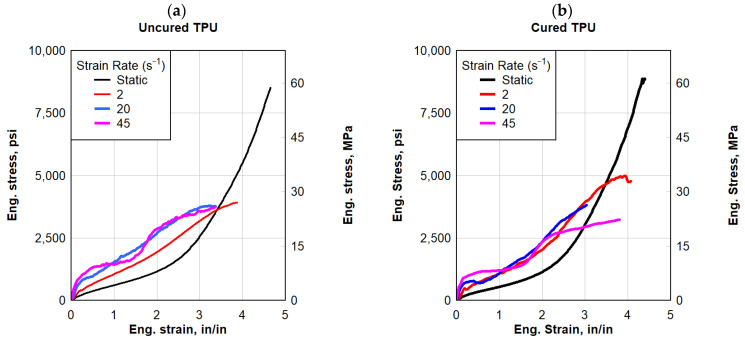
Eng. stress vs. eng. strain: (**a**) uncured TPU and (**b**) cured TPU.

**Figure 8 polymers-16-00730-f008:**
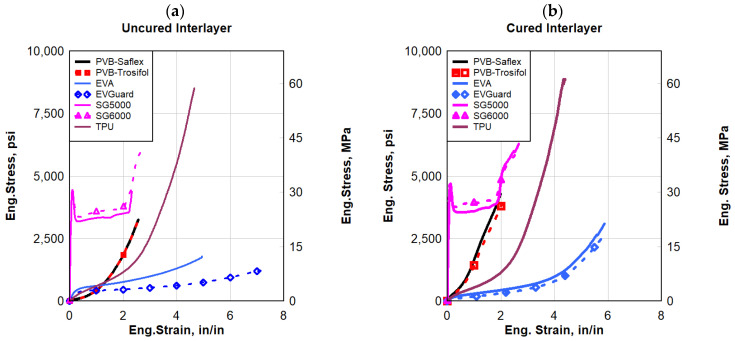
Eng. stress vs. eng. strain comparisons for quasi-static response: (**a**) uncured interlayer, and (**b**) cured interlayer.

**Figure 9 polymers-16-00730-f009:**
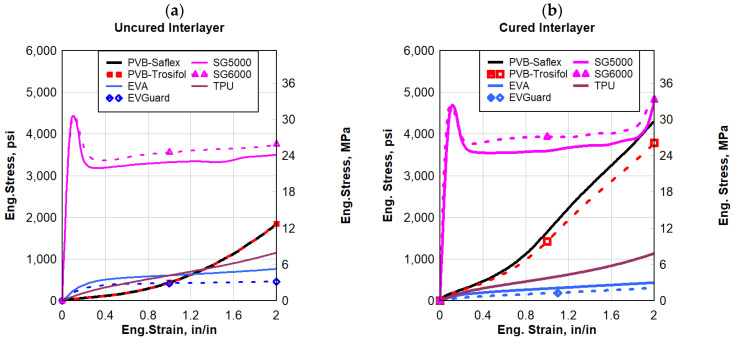
Quasi-static response comparisons of all polymers at an assumed design strain limit of 2 in/in; (**a**) uncured interlayer, and (**b**) cured interlayer.

**Figure 10 polymers-16-00730-f010:**
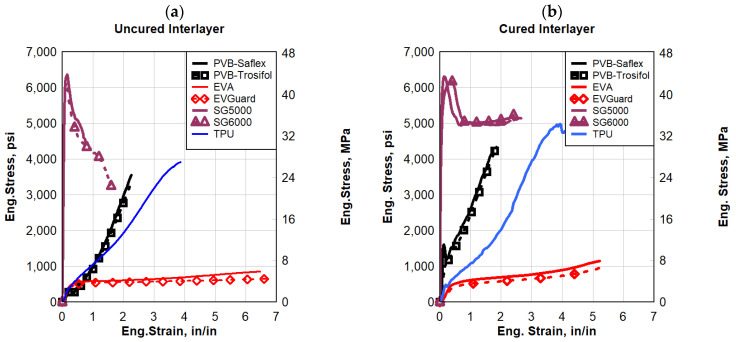
Eng. stress vs. eng. strain comparisons for a strain rate of 2 s^−1^: (**a**) uncured interlayer, and (**b**) cured interlayer.

**Figure 11 polymers-16-00730-f011:**
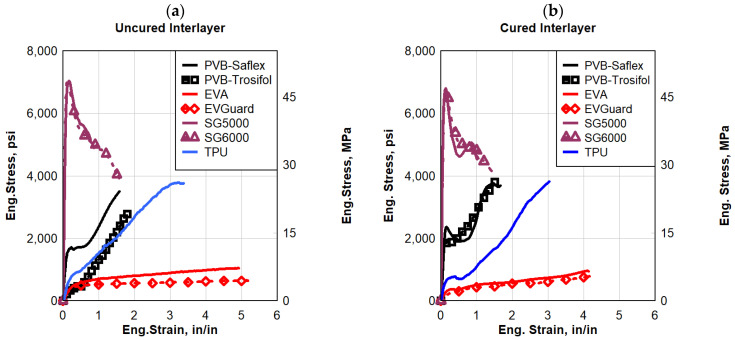
Eng. stress vs. eng. strain comparisons for strain rate 20 s^−1^: (**a**) uncured interlayer, and (**b**) cured interlayer.

**Figure 12 polymers-16-00730-f012:**
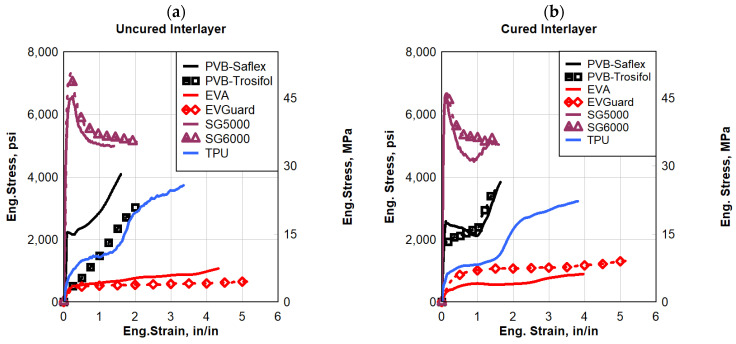
Eng. stress vs. eng. strain comparisons for a strain rate of 45 s^−1^; (**a**) uncured interlayer, and (**b**) cured interlayer.

**Figure 13 polymers-16-00730-f013:**
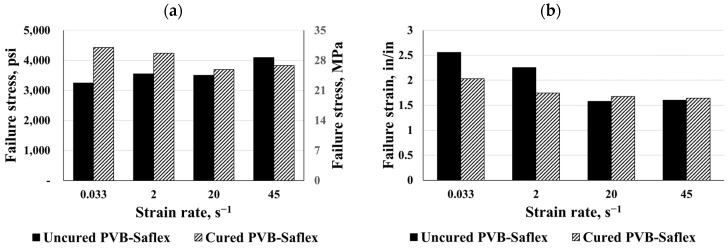
Comparisons for cured and uncured PVB-Saflex: (**a**) failure stress, (**b**) failure strain, (**c**) initial modulus, and (**d**) toughness.

**Figure 14 polymers-16-00730-f014:**
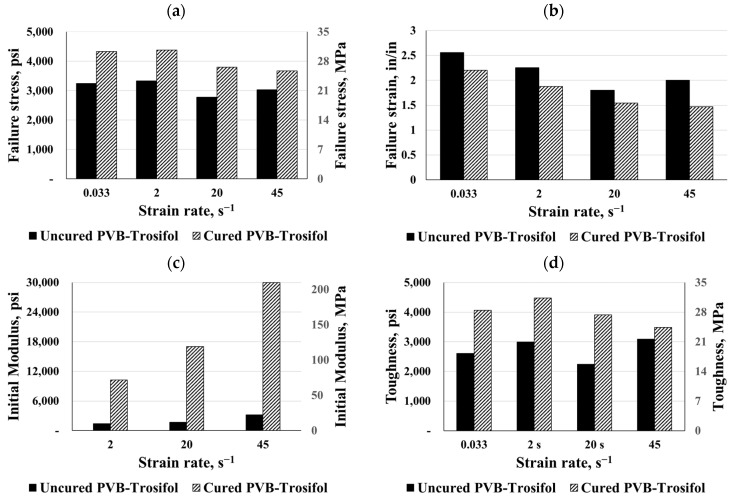
Comparisons for cured and uncured PVB-Trosifol: (**a**) failure stress, (**b**) failure strain, (**c**) initial modulus, and (**d**) toughness.

**Figure 15 polymers-16-00730-f015:**
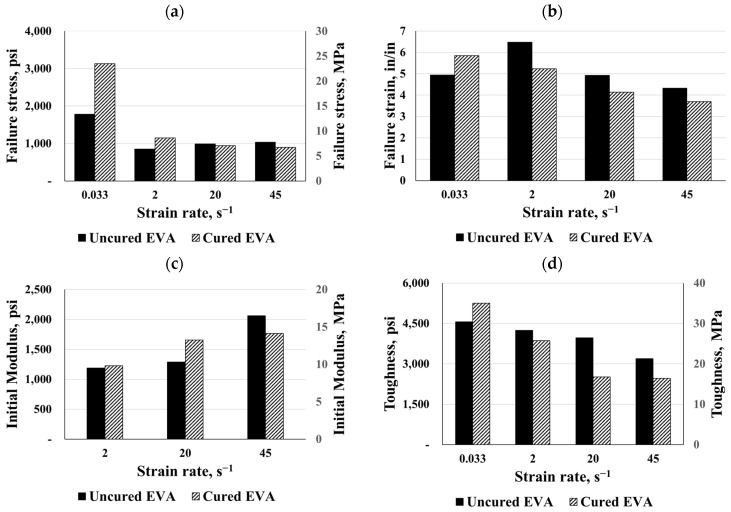
Comparisons for cured and uncured SE381-TF EVA: (**a**) failure stress, (**b**) failure strain, (**c**) initial modulus, and (**d**) toughness.

**Figure 16 polymers-16-00730-f016:**
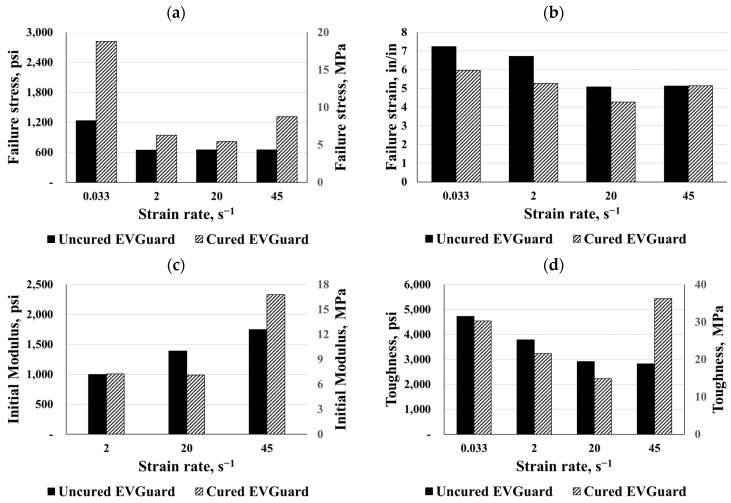
Comparisons for cured and uncured EVGuard EVA: (**a**) failure stress, (**b**) failure strain, (**c**) initial modulus, and (**d**) toughness.

**Figure 17 polymers-16-00730-f017:**
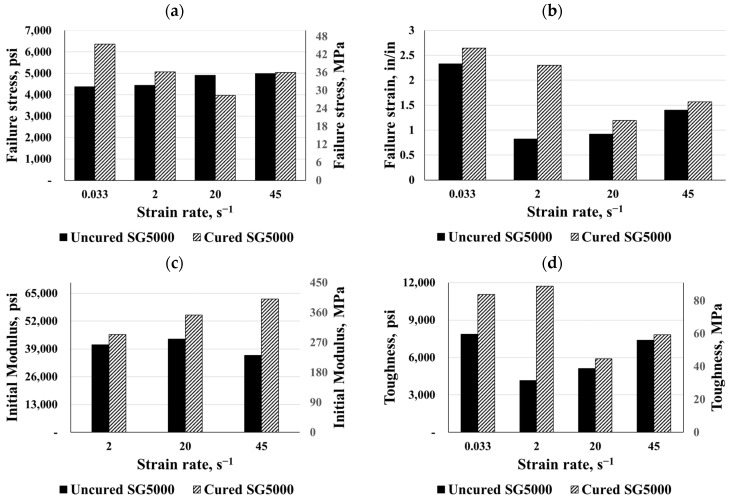
Comparisons for cured and uncured SG5000: (**a**) failure stress, (**b**) failure strain, (**c**) initial modulus, and (**d**) toughness.

**Figure 18 polymers-16-00730-f018:**
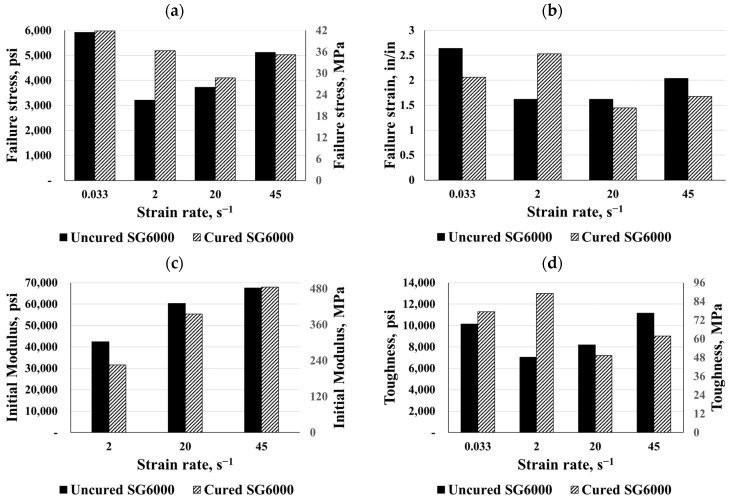
Comparisons for cured and uncured SG6000: (**a**) failure stress, (**b**) failure strain, (**c**) initial modulus, and (**d**) toughness.

**Figure 19 polymers-16-00730-f019:**
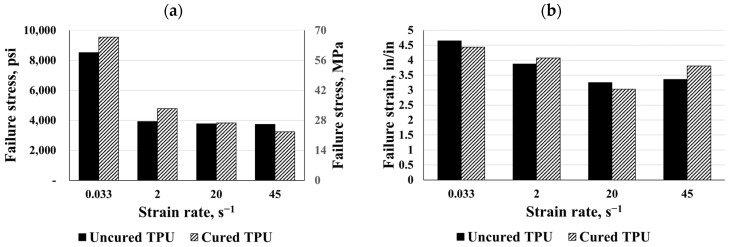
Comparisons for cured and uncured TPU: (**a**) failure stress, (**b**) failure strain, (**c**) initial modulus, and (**d**) toughness.

**Figure 20 polymers-16-00730-f020:**
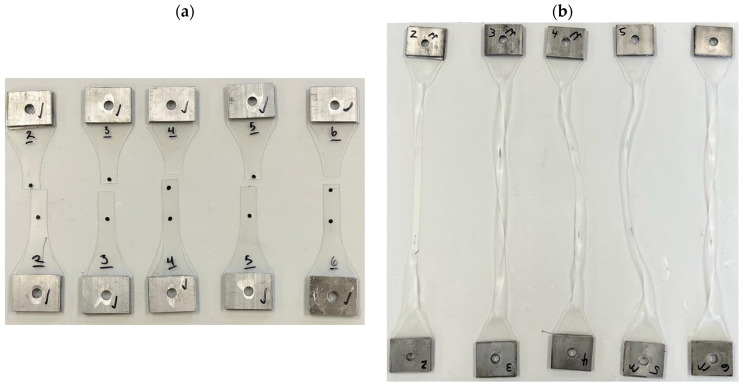
Specimens after testing: (**a**) PVB, (**b**) EVA, (**c**) SG, and (**d**) TPU.

**Table 1 polymers-16-00730-t001:** Polymer sheets were used in this study.

Material	Manufacturer/Product	Thickness in (mm)	Material State
**PVB**	Estman, Kingsport, TN, USA/Saflex Standard Clear	0.03 (0.762)	Uncured and cured
**PVB**	Kurary, Houston, TX, USA/Trosifol	0.03 (0.762)	Uncured and cured
**EVA**	SWM, Alpharetta, GA, USA/SE-381TF	0.03 (0.762)	Uncured and cured
**EVA**	Salem Fabrication Technologies, Winston-Salem, NC, USA/EVGuard	0.03 (0.762)	Uncured and cured
**SG5000**	Kurary, Houston, TX, USA/SG5000	0.035 (0.889)	Uncured and cured
**SG6000**	Kurary, Houston, TX, USA/SG6000	0.035 (0.889)	Uncured and cured
**TPU**	SWM, Alpharetta, GA, USA/Krystalflex PE 399	0.025 (0.635)	Uncured and cured

**Table 2 polymers-16-00730-t002:** Polymer testing results.

Polymer		Strain Rate	Yield Stress MPa	Yield Strain mm/mm	Failure Stress MPa	Failure Strain mm/mm	Young’s Modulus MPa	Toughness MPa-mm/mm
**PVB Saflex**	Uncured	Static	-	-	22.4	2.6	-	17.93
		2 s^−1^	2.5	0.28	24.5	2.25	10.7	22
		20 s^−1^	11.25	0.19	24.15	1.58	66.9	23.85
		45 s^−1^	16.5	0.14	28.3	1.6	190	30
	Cured	Static	-	-	30.5	2.0	-	26.3
		2 s^−1^	11	0.12	29.25	1.75	106	29.4
		20 s^−1^	16.5	0.15	25.5	1.7	154.4	29.9
		45 s^−1^	18.6	0.13	26.4	1.65	198	28.3
**PVB Trosifol**	Uncured	Static	-	-	22.4	2.56	-	18
		2 s^−1^	2.35	0.27	23	2.25	9.7	20.65
		20 s^−1^	1.7	0.1	20	1.87	12	15.5
		45 s^−1^	2.5	0.11	20.3	2.0	22.2	21.3
	Cured	Static	-	-	30	2.2	-	28
		2 s^−1^	8	0.165	30.2	1.9	70.8	30.9
		20 s^−1^	13	0.136	26.2	1.55	117	27
		45 s^−1^	13.4	0.06	25.3	1.5	206.6	24
**EVA SWM-SE**	Uncured	Static	3.5	0.44	12.3	4.93	10.3	31.5
		2 s^−1^	3.8	0.40	5.85	6.5	8.2	29.3
		20 s^−1^	3.7	0.38	7.2	4.9	9	27.4
		45 s^−1^	2.9	0.22	7.4	4.3	15	22
	Cured	Static	0.9	0.18	21.6	5.85	5.35	36.2
		2 s^−1^	3.6	0.43	7.9	5.25	8.5	26.6
		20 s^−1^	2.5	0.23	6.5	4.15	11.4	17.4
		45 s^−1^	2.75	0.25	6.2	3.8	12.2	17
**EVA EVGuard**	Uncured	Static	2.6	0.42	8.5	7.24	5.75	32.6
		2 s^−1^	3.3	0.44	4.5	6.7	6.9	26
		20 s^−1^	3.0	0.35	4.5	5.1	9.6	20.1
		45 s^−1^	2.9	0.24	4.5	5.1	12	19.5
	Cured	Static	-	-	19.4	6.0	-	31.3
		2 s^−1^	2.9	0.5	6.5	5.25	7	22.4
		20 s^−1^	1.8	0.29	5.6	4.3	6.8	15.4
		45 s^−1^	5.5	0.35	9	5.15	16	37.5
**SG5000**	Uncured	Static	30.6	0.1	30.15	2.3	316	54.25
		2 s^−1^	44	0.16	30.65	0.82	282	28.6
		20 s^−1^	49	0.14	33.8	0.91	301	35
		45 s^−1^	48	0.15	34.3	1.5	250	51
	Cured	Static	32.4	0.12	44	2.65	304	76.25
		2 s^−1^	43.7	0.13	35	2.3	315	80.8
		20 s^−1^	47.3	0.14	27.35	1.2	380	40.7
		45 s^−1^	46.2	0.12	34.8	1.57	430	54
**SG6000**	Uncured	Static	30.75	0.11	40.85	2.65	285	70
		2 s^−1^	42.5	0.14	22.1	1.62	293	48.5
		20 s^−1^	49.25	0.13	26	1.62	416	56.4
		45 s^−1^	50.85	0.15	35.1	2.0	466	77
	Cured	Static	32.2	0.09	41.2	2.1	365	77.75
		2 s^−1^	42.7	0.36	35.8	2.53	218	89.6
		20 s^−1^	47.3	0.14	28.3	1.45	382	50
		45 s^−1^	48.1	0.13	34.7	1.7	470	62.2
**TPU** **PE 399**	Uncured	Static	-	-	58.8	4.65	-	78.4
		2 s^−1^	2.8	0.2	27	3.9	16.3	53.7
		20 s^−1^	5.5	0.25	26	3.25	20.8	53
		45 s^−1^	5.8	0.15	25.7	3.35	30.8	52.7
	Cured	Static	-	-	66	4.4	-	78.5
		2 s^−1^	3.3	0.18	33	4	21.8	68.2
		20 s^−1^	4.7	0.17	26.4	3	25.7	39.5
		45 s^−1^	6.4	0.16	22.3	3.8	31.5	53.6

## Data Availability

Data are contained within the article.

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
