# Peer review of "The Influence of Strain Rate Behavior on Laminated Glass Interlayer Types for Cured and Uncured Polymers"

_polymers, 2024, doi:10.3390/polym16060730_

Round 1

Reviewer 1 Report

Comments and Suggestions for Authors

This work studied the quasi-static and dynamic response of cured and uncured for seven different materials in the recommended room temperature. The experimental results of this work are rich and well discussed, and the conclusions have good reference significance for research in related fields. This manuscript can be accepted after a minor revision.

My comment: Nowadays, there are numerous methods for theoretical simulation. Can the author consider using reliable theoretical simulation methods to verify the experimental results. This looks like it can better improve the quality of the manuscript.

Reviewer 2 Report

Comments and Suggestions for Authors

The work deals with the experimental investigation of some mechanical properties of some composite polymers glass samples widely used in practically applications especially the building domain. The investigation includes quasi-static and dynamic response of cured and uncured polymeric samples at temperatures belonging to the domain 21 to 32°C. The response of samples was evaluated under a quasi-static strain rate of 0.033 s-1 and compared to the results under a relatively higher strain rate 18 of 2 s-1.

            The experimental procedure is very well conducted and very well presented; the data are reliable and convincing; the figures are very suggestive and well selected; the data are well discussed. However in my opinion a simulation of experimental data with adequate mathematical model should bring more value to the paper. This suggestion is optional.   

            The paper is well written, the results and conclusions are correct, well presented, the references are of the actuality, reason for which I consider that the paper could be published as it is.

Reviewer 3 Report

Comments and Suggestions for Authors

Comments on the Quality of English Language

N/A.

Round 2

Reviewer 3 Report

Comments and Suggestions for Authors

Accept.

Comments on the Quality of English Language

Minor comments may be needed.